# Positive Organisational Arts-Based Youth Scholarship: Redressing Discourse on Danger, Disquiet, and Distress during COVID-19

**DOI:** 10.3390/ijerph18115655

**Published:** 2021-05-25

**Authors:** Ann Dadich, Katherine M. Boydell, Stephanie Habak, Chloe Watfern

**Affiliations:** 1School of Business, Western Sydney University, Locked Bag 1797, Penrith, NSW 2751, Australia; 2Black Dog Institute, University of New South Wales (UNSW), Sydney, NSW 2031, Australia; K.Boydell@blackdog.org.au (K.M.B.); S.Habak@blackdog.org.au (S.H.); Chloe.Watfern@unsw.edu.au (C.W.); 3Arts & Design, University of New South Wales (UNSW), Sydney, NSW 2052, Australia

**Keywords:** positive organisational scholarship in healthcare, arts-based research, young people, mental health, resilience, pandemics, COVID-19

## Abstract

This methodological article argues for the potential of positive organisational arts-based youth scholarship as a methodology to understand and promote positive experiences among young people. With reference to COVID-19, exemplars sourced from social media platforms and relevant organisations demonstrate the remarkable creative brilliance of young people. During these difficult times, young people used song, dance, storytelling, and art to express themselves, (re)connect with others, champion social change, and promote health and wellbeing. This article demonstrates the power of positive organisational arts-based youth scholarship to understand how young people use art to redress negativity via a positive lens of agency, peace, collectedness, and calm.

## 1. Introduction

COVID-19—the term that changed the world—how we live, work, learn, shop—particularly for toilet paper [1,2,3]—and socialise. Importantly, COVID-19 affects: how we interact; and relatedly, our personal and social wellbeing [4]. For instance, following recent national and international directives [5,6,7], COVID-19 shapes: who can interact with who; when they can do it; and how.

These and potentially other COVID-19-related changes can compromise wellbeing. This is particularly the case among individuals and communities who are isolated and/or experience poor health and/or mental health [8,9], including children and young people [10,11,12,13,14]. For instance, among young people, this precarious period of COVID-19 has been associated with loneliness, distress, anxiety, and post-traumatic stress disorder [15,16,17,18]. Furthermore, the tsunami of current and projected unemployment rates is expected to have considerable implications for young people, now and in the future [19,20,21,22]. This is because financial hardship can compromise mental health for approximately two years [23].

For many young people, COVID-19 has created a new norm—that of pecuniary pressure. This follows three key reasons. First, many are or were part of the sectors that are adversely affected by COVID-19—notably, ‘Dance and Musical Theatre; Live Production Services; Music; Screen and Media; [as well as] Visual Arts, Craft and Design’ [24,25]. To contain COVID-19, many governments closed, postponed, or cancelled the avenues these young people often relied on, including, but not limited to: theatres, concert venues, cinemas, galleries, festivals, and community events. This in turn ‘exposed the unsustainable price of creative and culture work’ [26]. Furthermore, although many had relied on alternative avenues, like hospitality, to supplement their income and stave off poverty, these are also limited [27,28]. Consider, for instance: the closure of cafés, restaurants, bistros, and bars; as well as the postponement or cancellation of national and international events that would have required hospitality staff, like the Olympic Games and the Australian Grand Prix. Consequently, recent North American data suggest some small businesses—like those representing ‘Arts, entertainment, and recreation’ and ‘Accommodation and food services’ might ‘never reopen’, post COVID-19 [29]. Second, even before COVID-19, the wellbeing of those who are part of the aforesaid sectors is often far from ideal [30]. These individuals are part of the gig economy, whereby freelance or contingent work is completed fractionally for different employers, sometimes simultaneously [31,32]. Given the precarious nature of this work, their rates of depression, anxiety, substance use, and suicidal ideation typically surpass national averages [33,34,35,36,37,38,39,40,41,42]. Many oscillate between variable periods of temporary-employment and unemployment [43,44,45]. Additionally, those engaged in the night-time-economy can be exposed to risky behaviours [46,47,48,49]; they also have reduced access to services, given their unconventional business hours. Third, for many young people, access to health insurance is contingent on employment. Thus, unemployment can make healthcare prohibitive, particularly for those who are part of the aforesaid sectors [50].

In addition to the aforesaid challenges, calls for ‘social [rather than physical] distancing’ [51,52,53] can compromise a key ingredient of wellbeing—namely, connectedness. For instance, within a sample of 932 adults in the United Kingdom, 36.8% of those who practiced social distancing due to COVID-19 experienced poor mental health, particularly those of younger age groups [54]. This can ‘present future health complications for the general public post-COVID-19 pandemic’ [54]. This might partly be because, for some young people, home is not the most safe or supportive environment—this is particularly the case for those who identify as lesbian, gay, bisexual, transgender, and/or intersex LGBTI, [55], and those with abusive family members, as abuse can be exacerbated by economic uncertainty and stress [56].

Despite this ‘global pandemic’ [57], creative brilliance happens, particularly among young people. Worldwide, many have demonstrated remarkable resourceful resilience. They have harnessed newfound opportunities to be creative and innovative, often artistically. For instance, rather than reflect or succumb to negative discourse, they have found ways to: express themselves; connect and/or reconnect with others; and address community needs that might otherwise be the responsibility of the government, not-for-profit, and/or private sectors [58]. Consider for instance, how young people creatively used social media to produce narratives of responsibility, fun, and resistance [59]. Similarly, others have: drawn ‘world[s] where kindness defeats COVID-19’ [60]; built ‘social movements and creative projects around a different vision for our planet’ [61]; raised community awareness through videography [62]; performed virtual concerts [63]; or lent their choreographic talents to public health campaigns [64]. Please refer to the following URLs:https://abcnews.go.com/GMA/Living/video/college-students-perform-virtual-concert-amid-coronavirus-69766015 (accessed on 29 March 2021)https://www.youtube.com/watch?v=Phd0Uktc204 (accessed on 29 March 2021)

There is much to learn from young people’s creative brilliance. For instance, despite their oft-cited frustrations, fears, and fragilities [65,66,67], what ignited these and other examples; how did the creative processes evolve; how did the experience influence the young people involved; and how did such creative brilliance influence others? Such scholarship can have theoretical and practical value. This is because the development of notable theory requires approaches that constructively provoke beliefs and assumptions, rather than merely identify gaps, issues, or problems within the knowledge-base, which itself is largely based on prevailing beliefs and assumptions [68,69]. By examining young people’s creative brilliance, there is opportunity to determine the conditions that help or hinder it—it can also encourage a respectfully critical consideration of the associated dark-sides. As Alvesson and Kärreman [70] argued:


*Empirical material can… facilitate and encourage critical reflection: to enhance our ability to challenge, rethink, and illustrate theory… To problematize means to… open up and to point out the need and possible directions for rethinking and developing the theory… It is the unanticipated and the unexpected—the things that puzzle the researcher—that are of particular interest… theory development is stimulated and facilitated through the selective interest of what does not work in an existing theory… encouraging interpretations that allow a productive and noncommonsensical understanding of ambiguous social reality.*


Relatedly, such scholarship can inform programs that aim to inspire creative brilliance among young people, or—given the positive association between art and health and/or mental health [71,72,73]—their wellbeing.

This focus on young people’s creative brilliance is not to moderate or worse still, discount concerns about those who are vulnerable, at-risk of harm, apathetic, or self-absorbed [74,75,76]. But rather, it serves to redress the relative wealth of discourse on danger, disquiet, and distress. Instead of adding to scholarship that speaks of young people’s problems or young people *as* a problem [77,78,79], it purposely shines a spotlight on their ‘ordinary magic in extraordinary times’ [80]. This positive lens reflects that of positive organisational scholarship (POS).

POS is an established methodology that goads scholars— *sensu lato*—to examine, understand, and ultimately promote phenomena that is life-giving and flourishing, like experiences that generate positive emotion and/or bolster resilience [81]. Undergirded by critical theory [82], it is not pollyannish or ignorantly blissful—nor is its expressed intention to incite change. Rather than ‘diagnos[e]… a… need for change… [to] establish… interventions’ [83], as per the intention of its related counterpart, appreciative inquiry [84], POS purposely recognises and aims to clarify how organisations—that is, groups of people who pursue a shared cause—enact virtuous practices and embody generative experiences, despite the typical challenges of organisational life, like limited resources, including funds, workforce capacity, time, or networks, among others. This aligns with critical theory, which largely questions, if not rejects the dominance of the deficit model. As Cameron and colleagues [85] asserted:


*We see strong similarities between the aims of critical theory and those of POS. The foundational statement for POS invites researchers to imagine another world in which almost all organizations are typified by appreciation, collaboration, virtuousness, vitality, and meaningfulness. Creating abundance and human well-being are key indicators of success.... Significant attention is given to what makes life worth living. Imagine that scholarly researchers emphasize theories of excellence, transcendence, positive deviance, extraordinary performance, and positive spirals of flourishing.*


Since its advent, POS has been extended into healthcare (POSH) to intentionally consider, make sense of, and raise the profile of those instances within organisational life—be they large-scale or modest—that exceed the expectation of those who deliver, manage, administer, or receive healthcare, *sensu lato* [86,87,88]. This has also involved the use of the methodology, video reflexive ethnography (POSH-VRE)—detail on this methodology can be sourced elsewhere [89,90,91,92,93].

Building on these methodologies, this article makes a case for positive organisational arts-based youth scholarship. It demonstrates how art can be used to understand and promote positive experiences during crises, among young people. Like POS, arts-based research is an established methodology with a demonstrated capacity to visibilise the abstract and the ephemeral—that which can be difficult to articulate and codify [94,95,96,97,98]. It is not necessarily undergirded by particular theories—but rather, arts-based research can be used to expand the mind. As Barone and Eisner noted [99]:


*Arts based research can provide the stuff that researchers can use to promote theoretical understanding of the traditional sort. Thus, out of connoisseurship come opportunities to generate conventional theory… theory is not limited to statements or to propositions; it is not necessarily fettered to the linguistic… theory is a structure for the promotion of understanding.*


However, arts-based research is not typically used with an expressed focus on that which is life-giving or generative. Positive organisational arts-based youth scholarship serves to turn the scholar’s gaze to these phenomena.

What makes positive organisational arts-based youth scholarship distinct from other methodologies is its capacity to purposely encourage scholars to examine constructive collective artistic efforts among young people. Although POS [81], arts-based research [97], and youth studies [100] are established scholarly approaches in their own right, positive organisational arts-based youth scholarship brings them together to explicitly recognise and learn from the creative brilliance of young people. Although previous research has considered young people’s experiences using arts-based methods—like, photovoice, ecomaps, and body mapping [101,102]—such research typically considers their experiences of disquiet or distress—like, anxiety or psychosis. This is not to suggest that arts-based research has not helped to reveal positive experiences among young people. However, this research was not typically intentional—that is, the researchers did not typically commence their research with the expressed objective to examine ‘individual and collective strengths (attributes and processes) and discovering how such strengths enable human flourishing (goodness, generativity, growth, and resilience)’ [103] or ‘dynamics that are typically described by words such as excellence, thriving, flourishing, abundance, resilience, or virtuousness’ [85]—as is the case in POS. For instance, in their systematic review of the use of arts-based methods to approach issues sensitive for youth and children, Tumanyan and Huuki found the methods were largely used to ‘(1) recognize and make visible previously invisible experiences, acts, voices and histories; (2) nurture change and transformation in the lives of the youth; and (3) allow exploring the more-than-human, more-than-present and less-than-conscious aspects in the lives of youth and children’ [104]. As such, the methods were not deliberately used to ‘rigorously seek… to understand what represents the best of the human condition’ [105]. Positive organisational arts-based youth scholarship requires the scholars to sharpen their focus on the positive dynamics of young people’s lives.

To demonstrate its potential, this article describes how positive organisational arts-based youth scholarship can be used to examine, make sense of, and clarify the ways in which some young people exceeded expectation, positively managing a global crisis—namely, COVID-19. This case is substantiated with reference to digital exemplars, sourced from social media platforms and relevant organisations.

The purpose of this article is not to present results—but rather, to draw on purposely selected exemplars to demonstrate the potential of positive organisational arts-based youth scholarship, the associated methodological challenges, and how these can be managed. As such, it is *not* presented as a conventional empirical article, complete with the standard sections on method and results—nor were the exemplars identified via a systematic review of all possible instances, for this was beyond the scope of the expressed aim of this article. Instead, and akin to other methodological articles [106,107,108,109], it presents an argument for a different way to examine, understand, and appreciate the creative brilliance of young people. This is achieved by drawing on, rather than analysing, purposely selected secondary data that serve as exemplars. In essence, the thesis of this article is twofold. First, many young people have the capacity to exercise agency and give voice to their experiences, particularly during times of adversity. And second, positive organisational arts-based youth scholarship represents one methodology to understand and ultimately raise the profile of their brilliance.

## 2. Positive Organisational Arts-Based Youth Scholarship: Exemplars

The power of the arts lies in their multifaceted and adaptable nature. They can act as a means of resistance and activism, and/or convey narratives of community connectedness and social cohesion. At a time when public discourse frequently portrays young people as ‘risk takers’, ‘a threat to themselves or others’, and more dangerously, the reason for COVID-19, being denoted as the ‘millennial bug’ [110], evidence of young people’s creative brilliance can help to oppose these misconceptions.

Some young people are disengaging from traditional media due to limited trust in the information being presented, suggesting that authorities fearmonger by ‘exaggerat[ing]… claims about coronavirus’ [111]. Despite consuming news more readily, young people are fact-checking more regularly, while turning to social media and other unconventional outlets for COVID-19 facts. Dissatisfied with sensationalist media coverage of COVID-19 [112], groups of young people have—of their own volition—initiated theatrical projects to raise their voice; demonstrate that their ‘views are not on the news’ [113]; and educate their communities by translating official health messages into creative forms that are readily accessible via social media [59]. For example, originating as a group formed in 2008 to help orphans and children of parents with the human immunodeficiency virus (HIV), the South African Ndlovu Youth Choir used their electrifying mix of vocals and dance moves to dismantle myths about the COVID-19 by performing and choreographing a song to educate young people about the importance of: practicing good hygiene habits; maintaining social distancing; and avoiding false rumours [64]. Importantly, their performance was streamed on YouTube with subtitles, enabling those with internet access to readily engage with their messages. Although COVID-19 halted their international tour, members of the Ndlovu Youth Choir took it upon themselves to spread a message of calm and hope to their community and wider fanbase. And the effects were palpable, with correspondence received from all around the world, praising the choir for its efforts. For instance, one fan thanked the choir for their online performance—he found it heartening as he struggled with depression, finding little hope during a period of uncertainty [114].

Similarly, the UNICEF (United Nations Children’s Fund) Nigeria Young Advocates Network recognised that much of the information about COVID-19 was not always accessible or suitable for children [115]. Despite the hastiness of media outlets to produce safety protocols to combat COVID-19, the content of these protocols typically targeted adults. Developed by a team of youth advocates, with support and funding from the European Union and UNICEF Nigeria, the UNICEF Nigeria Young Advocates Network harnessed the power of storytelling through digital media to create a child-friendly animation, providing information to children on how to stay safe and protect loved ones during COVID. Members of this group used ‘simple language’ and created ‘relatable characters’ to ‘help children grasp how to protect themselves and prevent the spread of Coronavirus’ [116]. Young people were thus able to recognise the continuous effort needed to curb the spread of COVID-19, taking it upon themselves to strengthen communication and raise awareness of the dangers of COVID-19 to both children and their families. Scouts in Lebanon, Romania, and Portugal also leveraged technology to create an informative video, raising awareness of sound hygiene practices during COVID-19, while producing their own version of a viral hand-washing dance [62]. Scientists are continually learning more about COVID-19, and health advice changes regularly with its ever-evolving status. Young people are rising to the challenge by choosing to stay informed. Rather than accepting the negative public discourse, some have instead used creative and innovative art forms to share practical and accessible information to help control the virus. Please refer to the following URL:https://www.voicesofyouth.org/blog/creating-covid-19-awareness-children-through-animation (accessed on 29 March 2021)

Since the advent of the lockdowns associated with COVID-19, many young people have reported feeling isolated and lonely, with a disproportionate number feeling sadder than their adult counterparts [12,117,118]. Upon exploring the adverse impacts of COVID-19 on unemployment, social dislocation, and mental health, dynamic modelling predicts a grim future for many young people, with an approximate 25% increase in suicide rates over the next five years as well as a surge in the demand for specialist mental health services [119].

Despite this ‘lockdown’ [25] generation’s ominous forecast, as it bears the scars of COVID-19 for decades to come, young people continue to enact hope via social movements fostered through creative outlets. Shouldering crisis after crisis, young people have rallied together around a common purpose: to raise awareness of the detrimental mental health effects of COVID-19 and increase cohesiveness within their generation. For instance, the Coronavirus Time Capsule—a global youth theatre collaboration—captured this unique moment in history: the arrival of the virus and the subsequent lockdown of young people, worldwide [120,121]. Company Three, a theatre company led by young people, aged 11 to 19 years, created the project to provide young people with a safe platform to: share their experiences of COVID-19; be listened to and understood; and feel supported. In this project, over 3000 young people shared personal stories, reflecting on the humour, creativity, and resilience of their peers in the face of the pandemic. Spanning 18 nations, including the United Kingdom, Kenya, and Thailand, young people explored their hopes and dreams for: their future; their family: and their mental health—opportunities to recognise and celebrate. As the initiative grew, many of the participating young people addressed mental health, reflecting on the actions they had taken for themselves and others to remain positive, calm, and happy during the pandemic. The reflections took the form of colourful affirmations and reminders, placed throughout their homes and communities for people to remember to continue to: care for each other; raise awareness about COVID-19 and how to contain it; and go the distance. Similarly, in Australia, teddy bears and rainbows began appearing in gardens, on windowsills, and on fences to create some ‘social-distancing magic’ for children during the global coronavirus pandemic [122]. Please refer to the following URL:https://www.youtube.com/watch?v=fRXT-PZXNGE (accessed on 29 March 2021)

The small, yet profound impact of practicing hope as a creative, collaborative work during adversity (albeit socially distanced), highlights young people’s infectious brilliance. As reflected by the exemplars presented in this article, social distancing is a shared effort, and acting together while maintaining distance is how many nations strive to flatten the COVID curve. Rather than viewing young people as catalysing the global pandemic, it is possible to learn from these exemplars in which many young people demonstrated solidarity in the face of divisive rhetoric [61].

Amid climate strikes and the black lives matter movement, young people’s right to protest, to engage in collective action, and to organise, were restricted during COVID-19 in unprecedented ways [123,124]. As traditional policies frequently overlook young people, youth organisations and social movements often provide the only space where young people can develop their civic identities and advocate for systemic change.

Since COVID-19 and the associated lockdowns, young people have embraced digital media to express their political stances in creative ways, claiming agency that might not have been afforded otherwise [125]. With digital media creation and accessible editing tools, many have expressed political views via videos, memes, and artworks. Consider 21-year-old Australian, Fin Spalding, who developed an online artivism campaign, ‘combining art and activism in support of LGBTI people’ during the COVID-19 lockdown [126]. Art has been commonly used to push boundaries and underscore the continual discrimination that many within the LGBTI community experience—similarly, Spalding’s campaign via Instagram opened an artistic avenue for self-expression, while highlighting the plight of individuals who are at risk for simply being who they are. Spalding, now a lead for Amnesty International’s LGBTIQ network, stresses the importance of listening to the voices and leadership of young people as ‘they are capable and powerful enough to change society and influence policy’ [127]. Similarly, Mohib Faizy—a 19-year-old student in Afghanistan—used the internet to distribute video-recordings of himself during COVID, reading children’s books. He conveyed compelling messages about humanity to hundreds of children with limited access to education. Creative digital engagement enabled Faizy to provide equitable access to content-rich books to young people, thereby empowering their own path to learning.

On a broader scale, climate strike activists led by Greta Thunberg have taken to the internet via the hashtag, #climatestrikeonline, to continue engaging young people in modern civic action. During COVID-19, the digitisation of activism offered young people a low-barrier way to exercise agency and engage with artivism [128]—that is, creative, poetic, and sensorial political action. The Global Climate Strike movement developed art kits, containing designs for signs and posters for online events, inspiration for street murals, strike music, as well as tips for taking (socially distanced) group photos to sustain momentum during the pandemic [129]. One innovation prompted by COVID-19 restrictions was a 24-h web-conference, showcasing young people across the world who discussed issues in their region, interspersed with activism-related activities [130]. Similarly, climate strikes that were scheduled to be held in approximately 3500 locations worldwide, adapted to COVID-19 restrictions. Thousands of young activists took to social media, calling on politicians to ‘fund our future—not gas’ [130]. Although some governments took advantage of this public health crisis to silence dissenting voices among young people [131], the creative and innovative responses of some young activists demonstrate their brilliant capacity to challenge the discourse around danger, disquiet, and distress by mobilising the masses. Please refer to the following URL:https://twitter.com/fff_digital/status/1249512415533957121 (accessed on 29 March 2021)

Collectively, the five exemplars presented in this article demonstrate the potential of positive organisational arts-based youth scholarship—these include: (1) the South African Ndlovu Youth Choir; (2) the UNICEF Nigeria Young Advocates Network; (3) the Coronavirus Time Capsule; (4) the digital creativity demonstrated by Fin Spalding and Mohib Faizy; and (5) the Global Climate Strike movement. Specifically, these exemplars were examined and critiqued with references to four lines of inquiry—namely: what happened that might be described as excellent, thriving, flourishing, abundant, resilient, virtuous, or brilliant; why; what influenced it, or might have; and what were the associated effects (see Table 1)? Responses to these questions were reported in a narrative form to demonstrate the application of positive organisational arts-based youth scholarship. As such, these exemplars are intended to guide the future use of this methodology, rather than prescribe how it should be used.

Evidently, a key limitation in this article is the reliance on secondary data. For instance: the exemplars were created for other purposes, and not for the purpose of this article; the data were limited to that sourced from the public domain, and thus might be incomplete; and research can be disembodied [132,133,134].

Despite this limitation, the five exemplars helpfully demonstrate the potential of positive organisational arts-based youth scholarship and what it can reveal about young people’s brilliant creative capacities. Guided by these exemplars, there is now considerable opportunity to trial the methodology with alternative forms and sources of data, including primary data collected via myriad art forms, like photovoice, ecomaps, and body mapping, among others [101,102]. There is also opportunity to involve young people as co-researchers in these efforts to raise the profile of their creative brilliance to different audiences—this might include refereed scholarly articles for academics, illustrative policy briefs for policymakers, as well as art exhibitions—be they on or offline—for the general public.

## 3. Discussion

Although young people might not be the most physically affected by COVID-19, its consequences have been far-reaching, exposing the frailties and inequalities of many Western economic and social systems. As noted by the European Youth Forum [131], young people are overrepresented in the hardest hit sectors of the economy, with one in six having ceased employment since the start of the crisis. As well as the strain on their financial situation, the pandemic has disrupted young people’s education, with temporary school closures affecting more than 91% of students worldwide—this equates to approximately 1.6 billion children and young people. Although many schools and teachers have adapted curricula to accommodate COVID-19, young people are at greater risk of social exclusion given the loss of support networks associated with the school environment.

This article argues that the methodology of positive organisational arts-based youth scholarship can be used to draw attention to and clarify positive experiences among young people during crisis. Specifically, it demonstrates how the arts have been used to understand and promote positive experiences during this global pandemic. Young people, worldwide, have challenged the harrowing statistics by demonstrating remarkable creative brilliance and resilience. For example, some have exercised agency by creatively promoting public health via choreographed dances and songs to communicate best hygiene practices for others within their community. Rather than remaining locked down, they chose to enact hope and directly address mental health via colourful affirmations and reminders, as well as teddy bears scattered in unassuming places. And instead of staying silent amidst global protests and movements, many young people adopted innovative and creative approaches to advocate for systemic change in the form of videos, street art, and digital media. By using the arts to depict their own lived experiences and phenomena that are critically important, many young people, worldwide, have fortified their strength and resilience, created meaning and purpose, (re)developed a sense of collective good, and built positive emotions for themselves and their community.

## 4. Conclusions

Using current digital exemplars, this article demonstrated how positive organisational arts-based youth scholarship can be used to examine, make sense of, and clarify the ways that some young people exceeded expectation. By positively using art, they redressed the negativity associated with COVID-19 via a positive lens of agency, peace, connectedness, and calm. There is no doubt that many young people can exercise agency and give voice to their experiences, particularly during adversity. Positive organisational arts-based youth scholarship represents one methodology to understand and ultimately raise the profile of their brilliance.

## Figures and Tables

**Table 1 ijerph-18-05655-t001:** Positive Organisational Arts-Based Youth Scholarship Lines of Inquiry.

Positive Organisational Arts-Based Youth Scholarship
What happened that might be described as excellent, thriving, flourishing, abundant, resilient, virtuous, or brilliant?
2. Why?
3. What influenced it, or might have?
4. What were the associated effects?

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
