# Peer review of "Positive Organisational Arts-Based Youth Scholarship: Redressing Discourse on Danger, Disquiet, and Distress during COVID-19"

_ijerph, 2021, doi:10.3390/ijerph18115655_

Round 1

Reviewer 1 Report

Dear authors,

thank you for providing me an updated image of arts-based youth scholarship during COVID-19. An article is well-written and contains a long list of references. Despite a vast number of exemplars, the manuscript does not convince me scientifically in its current form. My main concern is that the purpose of the article is not fully fulfilled; therefore, I would suggest the authors to analyze and classify the exemplars to provide the readers a coherent view on the topic and especially the method. In the following I suggest changes to the manuscript that I hope to improve the manuscript.

  • Title: Should you mention COVID-19 in the title since the whole text is linked to the pandemic?
  • The affiliation of CW is missing
  • Keywords: ´Knowledge translation´ is mentioned as a keyword but does not exist in the text – should be deleted
  • Consider using brackets less in Introduction since they challenge reading
  • If you use straight quotations, a page number is recommended to add, e.g. reference 26
  • Since this is a methodological article, a methodology should be introduced more in detail as well as compared to other similar methodologies: what makes the arts-based youth scholarship unique? How is it similar or different compared to other arts-based methodologies? Why is it limited or focused on youth? What are the limitations of the method? More critical and analytic investigation is needed in a methodological article and it should be provided as an own chapter.
  • The authors mention that POSH-VRE is an established methodology. If so, please refer also to other references, not only to your own citations.
  • Chapter 2 provides a wide list of exemplars of positive organizational arts-based youth scholarship. Many citations are Internet pages that is understandable based on the current topic. Many citations were also missing as Error! texts show. The methodology of the exemplars would be easier to understand if the authors analyze and classify the exemplars based on their methodology: what was done, how/form of method, what were the strengths and weaknesses of the method, possible outcome/implications etc. Also, to reflect the wide scale of exemplars, information could be collected to a Table. The authors should also provide more information on how they chose these exemplars among others; are they all that can be found or is there a systematic/unsystematic way to choose the exemplars?
  • Discussion in its current form mostly repeats what has been earlier said. The text should go deeper in the topic by providing information on e.g. how to apply the methods described in the article in practice, what is the role of researchers and others in applying the method, how the context, strengths, weaknesses and limitations of the methods should be taken into account and what should be done next to redress discourse on danger, disquiet, and distress during COVID-19.
  • Author contribution: Is SB SH?
  • References: The authors should be listed instead of using ´et al.´

Reviewer 2 Report

Reviewer 1 comments:

  1. Format: the English is correct but uses many parenthesis or inferred information that makes the understanding difficult to the reader. See lines 2-4 in paragraph 3 in Introduction, 2nd and 3th paragraph in page 2/12, for instance. For an external reader, that in many cases will not have English as mother tongue, I find it is clearer to widen the sentences to show the two meanings and not to write it in a so condensed way, which results ambivalent in meaning.
  2. Regarding structure and content:

Research problem:

I don’t see clearly stablish the research problem, the objective of research question that this research tries to clarify. Is it that young people is far away from typical media and that authors want to show it is not true by introducing different examples that they analyze through Positive Organizational Scholarship and Arts-based phenomena? Is it that covid-19 has contributed to eliminate arts-based opportunities and that young people has been able to counteract difficulties and offer creative arts-based actions that authors analyzed by POS?

Theoretical framework:

In the Introduction section, paragraph 5th page 3/12  is it presented the POS as a new perspective but there is missing the theoretical framework to describe and support it. There are several references related to it (84-88), but the information included in the introduction is not clear enough to have an idea of which is it consisting on, for a non-expert reader.

See for clarifying the foundations of POS:

Dutton, J. E., Glynn, M. A., & Spreitzer, G. (2005). Positive organizational scholarship. San Francisco.

Cameron, K. S., & Spreitzer, G. M. (Eds.). (2011). The Oxford handbook of positive organizational scholarship. Oxford University Press.

Cameron, K., & Dutton, J. (Eds.). (2003). Positive organizational scholarship: Foundations of a new discipline. Berrett-Koehler Publishers.

Cameron, K. S., & Caza, A. (2004). Introduction: Contributions to the discipline of positive organizational scholarship. American Behavioral Scientist47(6), 731-739.

It is the same case for arts-based methodologies, which foundations are missing.

Methology: although it is said that the work is descriptive, it needs a complete section that is missing. It is said the “reflective video analysis” is used, but it must be describe more deeply.

Results section: the section entitled 2. Positive Organisational Arts-Based Youth Scholarship: Exemplars seems to be the Results section. As the methodology has not been described, the section cannot be analyzed in terms of the items that should appear or not.

Discussion section: it is very week and short. In a discussion section the comparison with previous literature on young arts-based production would be expected, but is missing.

The second paragraph says “This article argues that the methodology of positive organisational arts-based youth scholarship can be used to draw attention to and clarify positive experiences among young people during crisis. Specifically, it demonstrates how the arts have been used to understand and promote positive experiences during this global pandemic.” But evidences to support it are not clearly stablished in the paper.

Therefore, my recommendation is that authors must reorganize the information to match a research article structure and content.

Round 2

Reviewer 1 Report

Dear authors,

thank you for the revised manuscript. You have worked hard based on reviewers´ comments and I can see vast improvements in the text. I have no further suggestions for the manuscript. I wish you all the best with your research. 

Reviewer 2 Report

Accept the new version, although the structure, as authors indicate, doesn't comply with the IMRD because of the type of reflection. Nonetheless, since it is explicitely indicated, I assume it has been made clearer and so it can be published.